# Leptin Stimulates Endometriosis Development in Mouse Models

**DOI:** 10.3390/biomedicines10092160

**Published:** 2022-09-01

**Authors:** Tae Hoon Kim, Nayoung Bae, Taeho Kim, Albert L. Hsu, Mark I. Hunter, Jung-Ho Shin, Jae-Wook Jeong

**Affiliations:** 1Department of Obstetrics, Gynecology & Reproductive Biology, Michigan State University, Grand Rapids, MI 49534, USA; 2Division of Reproductive Endocrinology, Department of Obstetrics and Gynecology, Guro Hospital, Korea University Medical Center, Seoul 02841, Korea; 3Department of Biomedical Engineering, Michigan State University, East Lansing, MI 48824, USA; 4Reproductive Medicine and Fertility Center, Department of Obstetrics, Gynecology & Women’s Health, University of Missouri School of Medicine, Columbia, MO 65202, USA; 5Department of Obstetrics, Gynecology & Women’s Health, University of Missouri School of Medicine, Columbia, MO 65202, USA

**Keywords:** obesity, leptin, endometriosis, animal model, db, ob, nanoparticle

## Abstract

Endometriosis is a chronic inflammatory condition in women, and obesity leads to an inflammatory condition that is directly involved in the etiology of endometriosis. However, observational studies have shown an inverse correlation between endometriosis and a low body mass index (BMI). Obesity does not protect against endometriosis, and on the contrary, an increased BMI may lead to more severe forms of the disease. To determine the effect of obesity on endometriosis, diet-induced and genetically engineered obese mouse models were integrated with endometriosis mouse models with fluorescence-tagged ectopic lesions. High-fat diet-induced obese mice revealed a significant increase in endometriosis development compared with regular-diet control mice. However, obese recipient mice with leptin deficiency and leptin receptor deficiency showed suppressed endometriosis development compared with control mice. Furthermore, donor uterine tissues with leptin deficiency and leptin receptor deficiency suppressed endometriosis development compared with control donor in control recipient mice. Importantly, we revealed that aberrant high levels of leptin concentration significantly increased endometriosis development compared with vehicle treatment group in control mice with normal body weight. Our results suggest that leptin and its receptor are critical for endometriosis development.

## 1. Introduction

Endometriosis is a common, benign gynecological disease that affects 6–10% of women of reproductive age, which is often associated with debilitating pelvic pain, dysmenorrhea and infertility [1,2]. It is defined by the presence of endometrial stroma and glands outside of the endometrial cavity [3]. America’s estimated yearly cost to diagnose and treat endometriosis exceeds USD 22 billion [4,5]. Current treatments include the surgical removal of the lesions or the induction of a hypoestrogenic state [1,6,7,8,9]. However, the recurrence of endometriosis after surgery or medical treatment is common, and the hypoestrogenic state is sub-optimal for women of reproductive age. Furthermore, current treatments are generally inefficacious [10,11].

Although the precise pathogenesis of endometriosis has yet to be fully understood, one widely accepted theory is the implantation of endometrial cells into the peritoneal cavity after retrograde menstrual flow through the fallopian tubes [12]. As fragments of endometrial tissue reach the peritoneal cavity, implant, and grow, ovarian steroid hormone signaling is altered to promote the growth of ectopic endometrial tissue [13]. The presence of ectopic tissue activates inflammatory processes that stimulate the growth of the lesions and promote angiogenesis. There is also increasing evidence that suggests that endometriosis is not limited to pelvic histologic manifestations but is rather a chronic, systemic and inflammatory process [14,15,16]. 

Obesity is considered a global pandemic and is more common than endometriosis, affecting an estimated 650 million adults in 2016, which is about 13% of the world’s adult population according to WHO Global Health Observatory Data [17,18]. It is also estimated that nearly 23% of women of childbearing age in the United States have obesity [19,20], which poses many risks including metabolic syndrome, sleep apnea, PCOS, and female infertility [21]. Obesity is both hyperestrogenic and inflammatory, which are separate pathways that may enable endometriosis lesions to thrive [22]. However, the relationship between endometriosis and obesity may be more complicated. Clinical studies on body mass index (BMI) and endometriosis have demonstrated mixed results. There is evidence of an inverse correlation between endometriosis and BMI, and obesity is often associated with a lower incidence of endometriosis [23,24,25,26]. In a meta-analysis of studies on BMI and endometriosis from 2019, lower BMI increases the relative risk of endometriosis. While low BMI shows an inverse correlation with endometriosis, obesity is not a protective factor for endometriosis [25]. Some studies suggest obese women suffer from endometriosis less frequently but that those with endometriosis may have more severe disease [27]. An observational, retrospective cohort study of 471 histologically confirmed endometriosis patients showed that patients with higher BMI are more likely to have superficial peritoneal lesions (odds ratio (OR) 1.070, 95% confidence interval (CI) 1.004–1.144; *p* = 0.044) [28]. Some have proposed that these mixed results on the effects of obesity on endometriosis may imply different mechanisms of obesity’s impact on disease development rather than a simple cause-and-effect relationship. As knowledge of the pathophysiology of endometriosis still remains to be elucidated, clinicians lack the guidelines to make proper recommendations for endometriosis patients regarding weight control. By identifying the mechanisms contributing to the early pathogenesis of endometriosis, we will be in a much better position to develop new and more effective therapies against endometriosis.

Leptin is a hormone predominantly made by adipose cells and enterocytes in the small intestine that helps to regulate energy balance by inhibiting hunger, which in turn diminishes fat storage in adipocytes. It is a product of the obese (*ob*) gene, which is a 16-kDa peptide hormone composed of 167 amino-acids [29,30]. In a person with a stable body weight, circulating levels of leptin are directly proportional to body fat mass [31]. Commonly known as the satiety hormone, it functions as a mediator in the long-term regulation of energy expenditure, reducing appetite in cases of weight excess [32]. In addition to its role in energy homeostasis, leptin signaling contributes to immune and angiogenic processes induced by IL-8, IL-6, TNF, and IL-1β and known to be crucial in the pathogenesis of endometriosis [33,34,35]. Increased levels of leptin in serum and peritoneal fluid specimens from women with endometriosis consistently have been reported since 2000 [31]. A meta-analysis of 25 studies with 2645 participants showed that leptin levels (weighted mean difference (WMD) = 4.45 mg/mL, 95% CI = 2.42–6.49, *p* < 0.01) and leptin/BMI ratio (WMD = 0.32 mg/mL, 95% CI = 0.23–0.42, *p* < 0.001) were significantly higher in women with endometriosis [36]. The concentration of leptin in the peritoneal fluid of endometriosis patients was found to be consistently high, while serum leptin concentration may be elevated or comparable [35,36]. Endometriosis is a complex, heterogeneous disease with a wide spectrum of clinical symptoms that can delay diagnosis. Much of its etiology and pathogenesis remain to be fully elucidated, making it difficult to design well-controlled experiments to study the growth and progression of the disease in humans. Therefore, understanding leptin and leptin receptor signaling pathways in the pathogenesis of endometriosis will provide new insights for better understanding the association between obesity and endometriosis and also for developing new strategies for treatment. 

In this study, we utilized obese mouse models driven by a high-fat diet, the leptin-deficient *ob/ob* mouse, and the leptin receptor deficient *db/db* mouse to determine the effect of obesity in endometriosis development. Our experiment results revealed the stimulation of endometriosis development in diet-induced obesity. However, obesity conditions with leptin deficiency or leptin receptor deficiency suppressed endometriosis development. Importantly, our studies revealed a critical role of leptin in the pathogenesis of endometriosis in mouse with normal body weight. 

## 2. Materials and Methods

### 2.1. Animals and Tissue Collection

Mice were maintained in a designated animal care facility according to Michigan State University’s Institutional Guidelines for the care and use of laboratory animals. All mouse procedures were approved by the Institutional Animal Care and Use Committee of Michigan State University. All housing and breeding took place in a designated animal care facility at Michigan State University with controlled humidity and temperature conditions and a 12-h light/dark cycle. Access to water and food (Envigo 8640 rodent diet, Madison, WI, USA) was ad libitum. For the diet-induced obese mouse model, female *Pgr^cre/+^Rosa26^mTmG^* mice were fed Michigan State University Vivarium food (regular) (17% of total calories from fat, Envigo, 8940, Madison, WI, ISA) or a high-fat diet (60% of total calories from fat, Researchdite, D12492i, New Brunswick, NJ ,USA) at 21 days of age until a designated time. To examine the effects of a high-fat diet, body weight was measured once a week. The leptin-deficient *ob*/*ob* (*Lep^ob^*; Strain #: 000632) and the leptin receptor-deficient *db*/*db* (*Lepr^db^*; Strain #: 000697) mice were purchased from the Jackson Laboratory (Bar Harbor, ME, USA).

### 2.2. Synthesis of Cy5.5 Dye-Doped Silica Nanoparticles

Tetraethyl orthosilicate (TEOS, 99%, Cat #86578), (3-aminopropyl)-triethoxysilane (APTES, 99%, Cat # 440140), Triethylamine (TEA, ≥99% Cat #T0886), and Cyanine5.5-NHS-ester (Cy5.5 NHS-ester, Cat # GEPA15502) were purchased from Sigma-Aldrich Chemicals (Atlanta, GA, USA). For nanoparticle synthesis, Cy5.5 dye-doped silica nanoparticles were prepared in two steps for silica sphere synthesis, followed by the covalent incorporation of Cy5.5 on the particles [37]. First, the Cy5.5-APTES solution was prepared. Specifically, amine-reactive, Cy5.5-NHS-ester (1 mg) was dissolved in 100 µL of dimethylsulfoxide (DMSO). Then, 1 μL of APTES was added, followed by the addition of 0.5 μL of TEA. The reaction mixture was stirred at room temperature in the dark. Next, for the silica synthesis, 1 μL of concentrated ammonia solution (28%), 8.5 mL of absolute ethanol, and 350 μL of TEOS were mixed and vigorously stirred at room temperature. In the following, the prepared Cy5.5-APTES dye solution (15 µL) was added to the reaction mixtures. After overnight reaction in the dark, the final nanoparticle solution was washed three times with ethanol by centrifugation and dispersed in DI water.

### 2.3. Ex Vivo Labeling with Cy5.5 Dye-Doped Silica Nanoparticles

To optimize the ex vivo labeling condition with Cy5.5 dye-doped silica nanoparticles in uterine tissue fragments, *Pgr^cre/+^Rosa26^mTmG^* donor mice were injected with 1 μg/mL of E2 daily for 3 days. Two uterine horns were collected 6 h after the last E2 injection. The uterine horn was opened longitudinally with scissors and then cut into small fragments of about 1 mm^3^ in a petri dish containing RPMI-1640 media (Gibco, Cat. #11835-030, Waltham, MA, USA) supplemented with 10% fetal bovine serum (FBS; Gibco, Cat # 16000044, Waltham, MA, USA). Small fragments were placed into 24 well plat (15 mg/well) and then incubated with 0, 5%, 10%, 20%, 40%, and 80% (*v*/*v*) Cy5.5 dye-doped silica nanoparticles for 3 h of incubation or 20% (*v*/*v*) Cy5.5 dye-doped silica nanoparticles for 0, 15, 30, 60, 120, and 180 min of incubation at 37 °C in 5% CO_2_. Free-floating nanoparticles were removed by washing 3 times with medium. 

### 2.4. Induction of Endometriosis

The induction of endometriosis was modified from previously described methods [38]. For autologous endometriosis induction, 8-week-old female mice were injected with 100 µL of 1 µg/mL estradiol (E2; Sigma Aldrich, Atlanta, GA, USA) in sesame oil daily for 3 days to synchronize uterine tissue. Six hours after the final injection, under anesthesia, a small midline abdominal incision was made, and one uterus was removed. One uterine horn was opened longitudinally with scissors and cut into small fragments of about 1 mm^3^ in a petri dish with a scalpel. Tissue fragments were placed into 500 µL PBS (Gibco, Cat. #10010031, Waltham, MA, USA), and tissue fragments were injected into the peritoneal cavity of the same mouse. For endometriosis induction with Cy5.5 dye-doped silica nanoparticles, E2 were injected into the donor mouse for 3 days. Six hours after the final injection, the mouse was euthanized, and the uterus was removed. The uterine tissue was cut into small fragments of about 1 mm^3^ in a petri dish with a scalpel. Tissue fragments were incubated with 20% (*v*/*v*) Cy5.5 dye-doped silica nanoparticles in a 24-well plate containing 500 µL RPMI-1640 medium (Gibco, Cat. #11835-030, Waltham, MA, USA) supplemented with 10% fetal bovine serum (FBS; Gibco, Cat # 16000044, Waltham, MA, USA). After 3 h incubation at 37 °C, 5% CO_2_, a small midline abdominal incision was made in the recipient mouse under anesthesia, and 45 mg of tissue fragments labeled with Cy5.5 dye-doped silica nanoparticles was injected into the peritoneal cavity of the recipient mouse. The abdominal incision was closed with sutures for the peritoneum and wound clips for the skin. After one month, the mice were euthanized, and GFP or Cy5.5-positive endometriotic lesions were counted and removed under a fluorescence-dissecting microscope. Volume of all endometriotic lesions were counted and calculated. Following H&E staining, histological signatures of endometriosis, including clearly demarcated endometrial glands, stroma, and fluid-filled cysts lined with single or multiple layers of epithelial cells were histologically confirmed from these tissues. 

### 2.5. The IVIS^®^ Spectrum In Vivo Imaging

Cy5.5 fluorescence signaling from the mice as well as incubated small uterine fragments was detected by IVIS^®^ Spectrum. Cy5.5 signaling intensity was quantified using Living Image^®^ software (PerkinElmer, Waltham, MA, USA).

### 2.6. Statistical Analysis

To assess statistical significance, we used one-way ANOVA followed by Tukey’s post hoc test for more than two group comparisons and Student t-test for two groups. A value of *p* < 0.05 was considered statistically significant. Statistical analyses were performed using the GraphPad Prism 9 (San Diego, CA, USA).

## 3. Results

### 3.1. Effects of Diet-Induced Obesity on Endometriosis Development

To evaluate the effects of diet-induced obesity on endometriosis development, high-fat diet-induced obese mice were surgically induced with endometriosis (Figure 1). Female *Pgr^ere/+^Rosa26^mTmG^* mice were fed a regular (17% of total calories from fat) or high-fat (60% of total calories from fat) diet at 3 weeks of age. The body weight was significantly higher in the high-fat diet group (19.25 ± 1.05 g) than in the regular diet group (15.55 ± 1.55 g) at 6 weeks of age (*p* < 0.05). At 3 months after the high-fat diet, the average body weight (37.50 ± 1.50 g) of the high-fat diet group had increased 1.9-fold compared with the regular diet group (19.50 ± 1.50 g). Endometriosis was surgically induced in both groups under their same diet conditions. Interestingly, the body weight was lower at 3 weeks after surgery and then recovered in the high-fat diet group, whereas the regular-diet group did not have any body weight changes (Figure 2A). One month after endometriosis induction, we observed GFP-positive endometriosis development in the high-fat diet group compared with the regular-diet group (Figure 2B). The high-fat diet group revealed significant (*p* < 0.05 and *p* < 0.001, respectively) increased number (13.00 ± 0.71) and weigh (183.78 ± 47.70 mg) of ectopic lesions compared with control (7.60 ± 0.51 and 56.98 ± 11.84 g, respectively) (Figure 2C). These results suggest that diet-induced obesity promotes the establishment and survival of endometriotic lesions. 

### 3.2. Ex Vivo Labeling with Cy5.5 Dye-Doped Silica Nanoparticles

The leptin-deficient *ob/ob* mouse and the leptin receptor deficient *db/db* mouse are the most commonly used genetically engineered animal models of obesity. Uncovering the pathophysiology of leptin signaling in endometriosis requires easy assays to distinguish endometriotic lesions from surrounding normal tissues. However, it takes a considerable amount of time to combine *Pgr^cre/+^Rosa26^mTmG^* mice and two obese mouse models. Therefore, we developed a mouse model of endometriosis by using ex vivo labeling with Cy5.5 dye-doped silica nanoparticles. To optimize the ex vivo labeling in uterine tissue fragments with Cy5.5 dye-doped silica nanoparticles, we first examined the concentration effects of the Cy5.5 dye-doped silica nanoparticles on ex vivo labeling. Synchronized uterine tissues treated with E2 after 3 days were collected from wild-type female mice. We divided an equal amount of uterine fragments (15 mg/well) [39] into each well and added 0, 5, 10, 20, 40, or 80% of the Cy5.5 dye-doped silica nanoparticles in 500 µL of RPMI-1640 complete media (10% FBS). After three hours incubation, we measured the intensity of Cy5.5 using IVIS after washing three times to clear unattached particles. Cy5.5 signaling (Total Radiant Efficiency [p/s]/[µW/cm^2^]) was significantly higher in 20% (4.36 × 10^7^ ± 5.45 × 10^6^; *p* < 0.05), 40% (6.65 × 10^7^ ± 1.26 × 10^7^; *p* < 0.05) and 80% (1.33 × 10^8^ ± 2.04 × 10^7^; *p* < 0.01) of Cy5.5 dye-doped silica nanoparticle concentrations compared to 0% (4.21 × 10^6^ ± 2.27 × 10^6^) (Figure 3A). To determine the optimal incubation time, we next incubated uterine tissue fragments labeled with 20% of Cy5.5 dye-doped silica nanoparticles for 0, 15, 30, 60, 120, and 180 min. After 120 min, the Cy5.5 signaling was significantly higher than with 0 min treatment (1.25 × 10^5^ ± 2.51 × 10^4^ and 3.58 × 10^5^ ± 4.05 × 10^4^; *p* < 0.01; Figure 3B). Furthermore, we confirmed uterine tissue fragments labeled with Cy5.5 dye-doped silica nanoparticles by histologic analysis. Cy5.5 signaling was detected in uterine tissue fragments that were treated with Cy5.5 dye-doped silica nanoparticles (Figure 3C) after 3 h incubation at 20% concentration. Therefore, we decided the optimal labeling condition was 20% concentration and 3 h incubation.

### 3.3. Effect of Obesity with Leptin-Deficiency and Leptin Receptor-Deficiency on Endometriosis Development in Mouse

*ob*/*ob* mice possess mutations in the leptin gene that lead to obesity. These mice display progressive food intake, insulin resistance, and moderately increased blood glucose levels. *Ob*/*ob* mice are phenotypically indistinguishable from their unaffected littermates at birth but gain weight rapidly throughout their lives. While the body weights of 8-week-old female heterozygote *ob*/+ mice was 21.62 ± 1.00 g, *ob*/*ob* mice revealed obesity with 44.13 ± 4.00 g body weight. To determine the effect of obesity with leptin deficiency on endometriosis, we surgically induced endometriosis in 8-week old *ob*/+ or *ob*/*ob* female recipients of donor uterine fragments from control *ob*/+ or *ob*/*ob* uteri. The donor tissues were labeled with Cy5.5 dye-doped silica nanoparticles. After one month from endometriosis induction, we examined the development of endometriosis by counting Cy5.5 signaling using IVIS. The number of ectopic lesions was significantly lower under the following conditions; 1) heterozygous donor and *ob*/*ob* recipient (1.17 ± 0.31; *p* < 0.05), 2) *ob*/*ob* donor and heterozygous recipient (1.33 ± 0.21 ; *p* < 0.01), and 3) *ob*/*ob* donor and *ob*/*ob* recipient (1.00 ± 0.26 ; *p* < 0.01), compared with heterozygous donor and heterozygous recipient (2.83 ± 0.40) (Figure 4). Our results suggest that leptin in ectopic lesions and body plays a critical role in endometriosis development. 

Leptin receptor (also known as LEP-R or OB-R) is a type I cytokine receptor, a protein that in humans is encoded by the LEPR gene. LEP-R functions as a receptor for the fat cell-specific hormone leptin. The *db*/*db* mouse is a genetically mutated mouse in which leptin receptors do not function properly. The *db*/*db* mouse is extremely obese while on control diet and has many of the metabolic defects (hyperphagia, hyperglycemia, hyperinsulinemia, and infertility) found in *ob*/*ob* mouse. The body weights of 8-week-old female *db*/*db* mice (20.14 ± 1.47 g) were significantly higher compared with heterozygote *db*/+ mice (40.87 ± 0.72 g). We surgically induced endometriosis in 8-week-old *db*/+ or *db*/*db* female recipients of donor uterine fragments from control *db*/+ or *db*/*db* to determine the effects of obesity with leptin receptor deficiency on endometriosis. The number of ectopic lesions was also significantly lower in 1) heterozygous donor and *db*/*db* recipient (0.67 ± 0.33; *p* < 0.05), 2) *db*/*db* donor and heterozygous recipient (1.33 ± 0.33; *p* < 0.01), and 3) *db*/*db* donor and *db*/*db* recipient (1.00 ± 0.00; *p* < 0.05) compared with heterozygous donor and heterozygous recipient (3.33 ± 0.67) (Figure 5). Our results demonstrated that obesity with leptin deficiency or leptin receptor deficiency does not stimulate endometriosis development.

### 3.4. The Effect of Leptin Signaling on Endometriosis Development

Our results from the *ob*/*ob* and *db*/*db* mice suggest a critical role of leptin signaling in endometriosis develpment. To determine the effect of leptin in endometriosis development, 8-week-old *Pgr^ere/+^ Rosa26^mTmG^* mice with endometriosis were administered vehicle or 15 mg/kg concentration of leptin for two weeks (three intraperitoneal injections per week). After vehicle or leptin treatment, the leptin-treated group showed a significant increase in the number of ectopic sites (7.33 ± 0.33) over the vehicle-treated group (5.00 ± 0.58; *p* < 0.05). In addition, the weights of the ectopic sites were also significantly higher in the leptin-treated group (60.80 ± 4.88 mg) compared with the vehicle-treated group (21.03 ± 4.35 mg; *p* < 0.01) (Figure 6). Our study demonstrates that an aberrant higher level of leptin stimulates endometriosis development in mice with normal body weight.

## 4. Discussion

The literature on the relationship between obesity and endometriosis is controversial. Many studies have demonstrated that an inverse relationship exists between BMI and the risk of developing endometriosis. However, some studies have reported more severe stages of endometriosis in obese women, implying that the association of obesity and endometriosis may be complicated [21,22]. In this study, we describe a possible mechanism for a relationship between endometriosis and obesity using different obesity mouse models. We utilized three distinct obesity mouse models to examine the importance and pathophysiological effect of obesity in endometriosis: (1) high-fat diet-induced obesity; (2) obesity with leptin deficiency (*ob*/*ob* mice); and (3) leptin receptor deficiency (*db*/*db* mice).

Several mouse models have been used to study endometriosis [40]. One widely used model introduces excised human endometrial fragments into the peritoneum of immunocompromised mice but is limited by the lack of a normal immune system, which is thought to be important in endometriosis pathophysiology [40,41,42]. Also widely used is the mouse model of induced endometriosis, especially for studying how the immune system [43], hormones [44,45], and environmental factors [46,47] affect endometriosis. Many transgenic mice are also available with specific genes either eliminated or overexpressed to study specific pathways in the development and progression of endometriosis [40]. However, current mouse models of endometriosis that involve ovariectomy and E2 treatment are impractical for studying physiological functions that require natural fluctuations in ovarian steroid hormones, such as fertility. Understanding the pathophysiology of endometriosis with mouse models requires easy assays to distinguish endometriotic lesions from surrounding normal tissues. With this in mind, we developed a mouse model of endometriosis based on *mTmG* reporters [38]. The expression of the *Luc* gene in this reporter mice is blocked by a loxP-flanked stop cassette until Cre-mediated excision [48]. *mTmG* mice have a double-fluorescent Cre reporter and express membrane-targeted tandem dimer Tomato (mT) prior to Cre-mediated excision and membrane-targeted green fluorescent proteins (mG) after excision [49]. In *Pgr**^cre^**^/^**^+^**Rosa26^Luc/mTmG^* mice, *Pgr*-positive uterine cells express luciferase and mG, and *Pgr*-negative cells express mT. However, we have to combine *Pgr^cre/+^Rosa26^mTmG^* mice and two obese mouse models to use this fluorescence reporter system, which takes a considerable amount of time. Therefore, we developed an ex vivo method of labeling donor uterine tissues with Cy5.5 dye-doped silica nanoparticles. Donor uterine fragments established endometriotic lesions in recipient mice without immune rejection. This endometriosis model closely mirrors human endometriosis and employs a Cy5.5 fluorescence reporter that allowed us to make quantitative assessments of the endometriotic lesions in these mice more accurately than in prior models. It also enables the noninvasive visualization of in vivo and real-time endometriotic lesions by fluorescence imaging and can be combined with obese mice or knock-out mice.

In this study, we used several mouse models to investigate how obesity affects the development of endometriotic lesions. First, we compared how increased body weight affects the establishment of endometriosis compared with normal weight. In the high-fat diet-induced obese group, the number and weight of endometriotic lesions were significantly higher compared with the regular-diet group with normal body weight. This result matches the previous reports that obesity does not protect against endometriosis and that, on the contrary, an increased BMI may lead to more severe forms of the disease. Interestingly, the body weights of the mice in the high-fat diet group decreased 3 weeks after the induction of endometriosis and later recovered. To explain this reduction in weight, one can argue that the surgery itself may have been a stress factor for the mice. However, it should be noted that the regular diet group did not share the same trend in body weight. Similar observations were also reported in another animal study in which decreases in body weight and total body fat content in mice were observed after the surgical induction of endometriosis [50]. Goetz et al. propose that metabolic disruption induced by endometriosis may be the cause of decreased body weight [50]. Some clinicians attribute the low BMI of endometriosis patients to chronic pelvic pain and inflammation that influences women’s appetite, eating patterns, and long-term calorie intake [22,51,52]. In fact, more severe stages of endometriosis seem to be associated with a lower BMI [27,53]. High levels of leptin also take part in reducing appetite and regulating energy intake. Thus, findings from the present study may provide further insight into leptin’s part in the inverse association of low BMI and endometriosis.

We used two genetically engineered obesity mouse models, *ob*/*ob* and *db*/*db* mice, to determine the effect of obesity without leptin and its receptor in endometriosis development. Both *ob*/*ob* and *db*/*db* mice represent animal models that eat excessively due to mutations in the gene responsible for the production of leptin and its receptor, respectively, and become profoundly obese. Surprisingly, both mice revealed significant decreases in endometriotic lesion development compared with control group. Our results suggest a clue regarding why we do not observe strong correlations between BMI and endometriosis. Recent evidence highlights the multiple links between endometriosis and systemic diseases with immunologic and metabolic changes and has identified comorbidities associated with endometriosis such as autoimmune diseases, cancers, and cardiovascular disease [54,55]. Therefore, studies targeting biological substances that specifically function in both adipose tissue metabolism and endometriosis could help better understand the observed relationship between adiposity and the establishment or behavior of endometriosis [23].

Leptin is a cytokine produced mainly by adipocytes, which not only plays a role in energy balance and homeostasis but also in pro-inflammatory and angiogenetic functions, which may be fundamental in the pathogenesis of endometriosis [31,35,36]. Leptin’s role in the development of endometriosis is not clearly understood, but increased levels of leptin in peritoneal fluid of endometriosis patients have been consistently reported [31,34,56,57,58]. Understanding this association between leptin, diet, and endometriosis may provide clues for clinicians on how to advise these patients and develop new strategies for this treatment-refractory and debilitating disease. In addition to its role in energy homeostasis and signaling, leptin is known to be involved in the mitogenic activity of epithelial cells [30] and in neoangiogenic factors [14]. Higher levels of leptin significantly enhance mitogenic activity in cultured eutopic and ectopic endometrial stromal cells as well [59]. Based on leptin’s angiogenic and mitogenic activity on the endometrium and its ability to modulate the immune response through receptors in T-cells (CD4), increased body weight and increased levels of leptin may promote endometriotic lesion establishment and progression [30,60].

To investigate leptin’s contribution to the pathogenesis of endometriosis, we designed and demonstrated how decreased leptin signaling negatively impacts the induction of endometriosis in a mouse model. In both leptin-deficient and leptin receptor-deficient groups, the disruption of leptin signaling resulted in the decreased activity of endometriotic lesions. Additionally, rescuing leptin signaling by treatment with leptin restored the disease activity and even affected the progression of already existing endometriosis at higher doses. In a previous study, it was demonstrated that the disruption of leptin signaling by the injection of a pegylated leptin peptide receptor antagonist impairs the establishment of endometriosis-like lesions [61]. Increasing peritoneal leptin levels are inversely proportional to disease extent, implicating leptin’s potential role in the early establishment of endometriotic lesions [61]. In the same study, the leptin receptor-deficient mouse also demonstrated similar impairments in endometriosis development as those treated with pegylated leptin peptide receptor antagonists [61]. Likewise, leptin has also been shown to promote the migration and invasion of endometriotic cells in a dose-dependent manner, which is in line with the present study’s findings [62]. Leptin is known to reflect adiposity, and as paradoxical as it may seem, there is an inverse relationship between BMI and the risk of endometriosis. Understanding the changes in body weight as a phenomenon caused by metabolic disruption of leptin signaling and development of endometriosis resolves this conundrum. This explains the elevated levels of leptin observed in earlier stages of endometriosis compared with more advanced stages of disease [31,57,58]. High levels of peritoneal fluid leptin have been consistently associated with endometriosis in humans [31,34,56,57,58]. If the pathophysiologic mechanisms of endometriotic lesion formation in mouse models could be translated to human biology, disruption of leptin signaling may provide a novel therapeutic target for endometriosis. Clinically, there are some data that women with endometriosis have mitochondrial dysfunction {Hsu, 2015 #1611}, which may also involve leptin signaling. For women with chronic pelvic pain, painful periods, pain with intercourse, and painful bowel movements during their periods, health care providers should still consider the possibility of endometriosis, even in women with a high BMI. Finally, there is increasing evidence that there may be different clinical or biochemical subtypes of endometriosis; we speculate that leptin signaling may play a role in the pathogenesis of at least some endometriosis subtypes.

In summary, we demonstrate the role of leptin and leptin signaling in the pathogenesis of endometriosis in an experimental mouse model. Further research should be targeted toward discovering the exact mechanisms in which body weight and obesity affects leptin signaling to participate in the formation and propagation of endometriotic lesions. The disruption of leptin signaling hinders the induction of endometriosis in mouse models. This implies leptin’s potential role in the development and progression of endometriotic lesions and targets for possible therapeutic interventions.

## Figures and Tables

**Figure 1 biomedicines-10-02160-f001:**
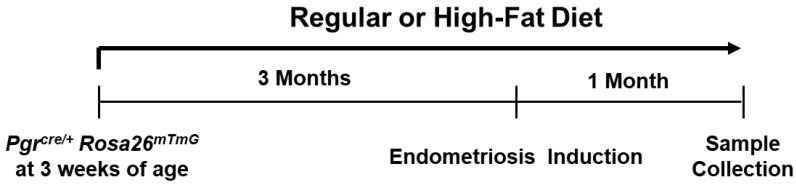
Experimental design to evaluate the effects of diet-induced obesity on endometriosis. Female *Pgr^cre/+^Rosa26^mTmG^* mice were feed a high-fat or regular diet to induce obesity at 21 days of age. Three months after obesity induction, endometriosis was surgically induced. The effect of diet-induced obesity on endometriosis was evaluated by examining number and weigh of ectopic lesions.

**Figure 2 biomedicines-10-02160-f002:**
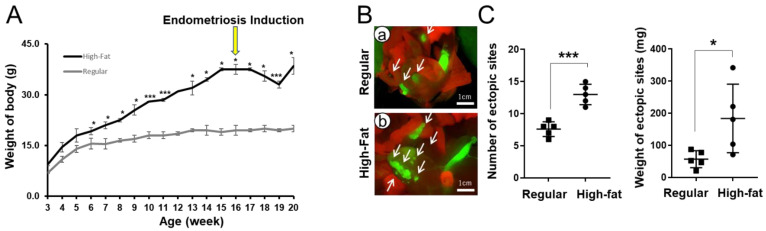
Effects of diet-induced obesity on endometriosis. (**A**) Changes in body weight in female *Pgr^cre/+^Rosa26^mTmG^* mice during the high-fat and regular diets. Female *Pgr^ere/+^Rosa26^mTmG^* mice were fed a regular or high-fat diet at 3 weeks of age. Body weight was measured once a week. Yellow arrow indicates the time of endometriosis induction. (**B**) Representative images of GFP-positive endometriotic lesions in high-fat (**b**) and regular (**a**) diet groups. Endometriosis was surgically induced in the regular and high-fat dieted mice. Endometriotic lesions were visualized under stereo microscope with fluorescence. (**C**) The quantification of number and weight of ectopic lesions were significantly increased in high-fat diet mice compared with regular diet mice. Data are represented as mean ± SEM. * *p* < 0.05 and *** *p* < 0.001.

**Figure 3 biomedicines-10-02160-f003:**
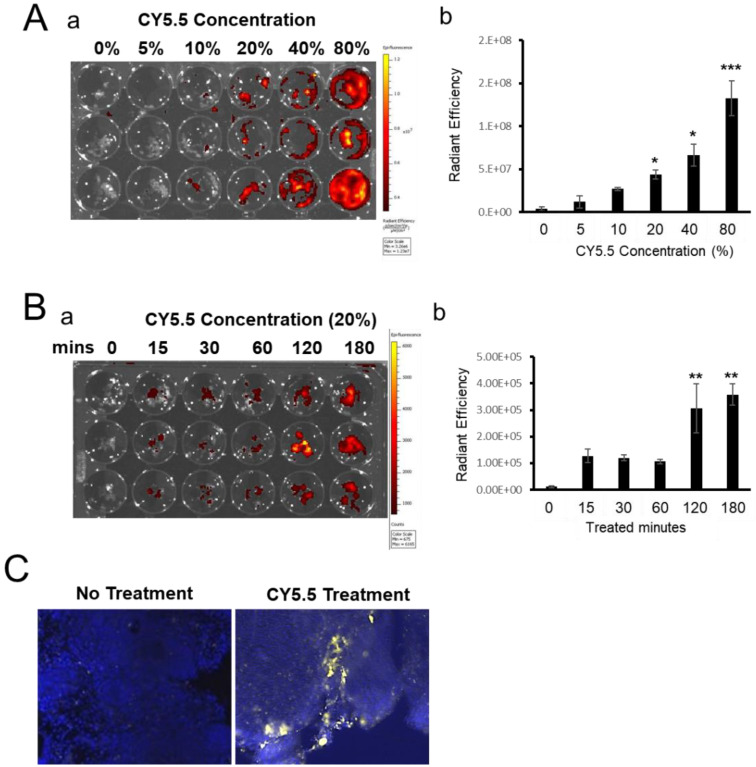
Optimization of ex vivo labeling condition with Cy5.5 dye-doped silica nanoparticles in uterine tissue fragments. (**A**) Representative images (**a**) and quantification (**b**) of Cy5.5 labeling at different concentrations of Cy5.5 dye-doped silica nanoparticles. Uterine tissue fragments were incubated with 0, 5, 10, 20, 40, 80% of Cy5.5 dye-doped silica nanoparticles for 3 h. The intensity of Cy5.5 was examined using IVIS spectrum. (**B**) Representative images (**a**) and quantification (**b**) of Cy5.5 labeling on different incubation times of Cy5.5 dye-doped silica nanoparticles. Uterine tissue fragments were incubated with 20% (*v*/*v*) Cy5.5 dye-doped silica nanoparticles for 0, 15, 30, 60, 120, 180 min. (**C**). Representative fluorescence images of Cy5.5 in uterine tissue without (a) and with (b) Cy5.5 dye-doped silica nanoparticles. Yellow fluorescent protein indicates Cy5.5 expression, and nuclei were counterstained with DAPI. Data are represented as mean ± SEM. * *p* < 0.05, ** *p* < 0.01, and *** *p* < 0.001.

**Figure 4 biomedicines-10-02160-f004:**
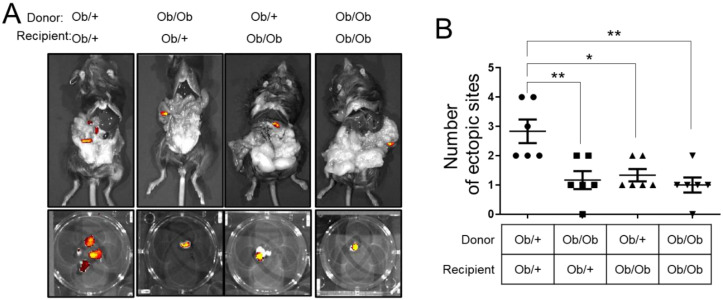
Effects of obesity with leptin deficiency on endometriosis. (**A**) Representative Cy5.5 signaling of *ob*/+ and *ob*/*ob* recipient mice with *ob*/+ and *ob*/*ob* endometriotic lesions. Endometriosis was surgically induced in 8-week-old *ob*/+ or *ob*/*ob* female recipients of donor uterine fragments from control *ob*/+ or *ob*/*ob* uteri labeled with Cy5.5 dye-doped silica nanoparticles. (**B**) Quantification of data for the number of ectopic sites were significantly lower in different donor and recipient condition compared with heterozygous donor and heterozygous recipient. Data are represented as mean ± SEM. * *p* < 0.05 and ** *p* < 0.01.

**Figure 5 biomedicines-10-02160-f005:**
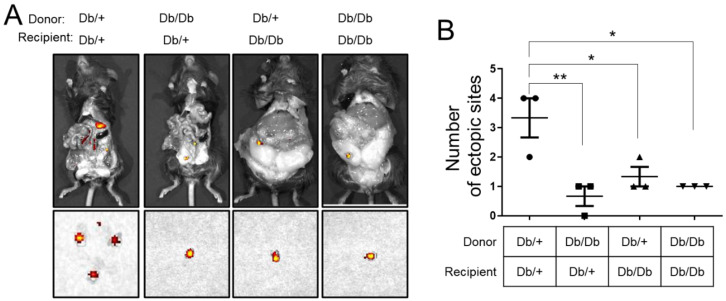
Effects of obesity with leptin receptor deficiency on endometriosis. (**A**) Representative Cy5.5 signaling of *db*/+ and *db*/*db* recipient mice with *db*/+ and *db*/*db* endometriotic lesions. Endometriosis was surgically induced in 8-week-old *db*/+ or *db*/*db* female recipients of donor uterine fragments from control *db*/+ or *db*/*db* uteri labeled with Cy5.5 dye-doped silica nanoparticles. (**B**) Quantification of data for the number of ectopic sites were significantly lower in different donor and recipient condition compared with heterozygous donor and heterozygous recipient. Data are represented as mean ± SEM. * *p* < 0.05 and ** *p* < 0.01.

**Figure 6 biomedicines-10-02160-f006:**
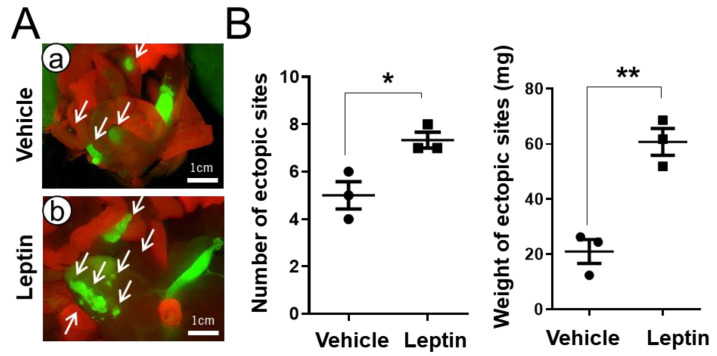
The effect of leptin treatment on endometriosis development. (**A**) Representative images of GFP-positive endometriotic lesions in *Pgr^cre/+^ Rosa26^mTmG^* mice treated with vehicle (**a**) or leptin (**b**) after endometriosis induction. *Pgr^cre/+^ Rosa26^mTmG^* mice were induced with endometriosis and then were treated with vehicle or leptin for 2 weeks (3 times intraperitoneal injection per week). Endometriotic lesions were visualized by GFP and arrows indicate ectopic sites. (**B**) The quantification of data for number and weight of ectopic sites were significantly higher in leptin-treated endometriosis mice compared with vehicle-treated endometriosis mice. Data are represented as mean ± SEM. * *p* < 0.05 and ** *p* < 0.01. Arrows indicate ectopic sites.

## Data Availability

Not applicable.

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
