# Peer review of "Leptin Stimulates Endometriosis Development in Mouse Models"

_biomedicines, 2022, doi:10.3390/biomedicines10092160_

Round 1

Reviewer 1 Report

The authors aimed at testing the role of leptin in the development of endometriosis using different mouse models. Additionally, a new and elegant approach was used to label the donor tissues. Few comments still remain :

- statistics: student t-test can only be used when assumption of normality has been checked. Did the authors check for normality? 

- to compare the number of lesions between groups, it is important to know how many tissues where initially injected; as such, the 'take rate' could be determined, defined as ration of lesions compared to the number of tissues injected. 

- the authors provide a new method to increase visibility of the tissues in the recipient mice by either a genetically modified mice (mTmG) or by labeling the tissues with cy5 containing nanoparticles. However, did the authors tested whether it was indeed easier to find the lesions (eg by comparing tissue retrieval with and without the aid of fluorescence)?

In addition, the cy5 labelled tissues are recovered one month later. However, the authors did not provide evidence of how long the nanoparticles or cy5 stain is able to remain over a period of time. Did the author perform longitudinal experiments to investigate the effect of time on the cy5 labeling?

- can cy5 labeling be readily analysed in ob/ob recipients with regard to adipose tissue? 

- the analysis of endometriosis in the different types of mice is rather limited. Were other aspects of the pathogenesis investigated as well (eg neuroangiogenesis, inflammation, pain behaviour)? 

Author Response

Reviewer #1

The authors aimed at testing the role of leptin in the development of endometriosis using different mouse models. Additionally, a new and elegant approach was used to label the donor tissues. Few comments still remain :

We are pleased with the positive comment.

C1. Statistics: student t-test can only be used when assumption of normality has been checked. Did the authors check for normality?

R1. The reviewer is correct. Yes, we have checked normality of the populations.

C2. To compare the number of lesions between groups, it is important to know how many tissues where initially injected; as such, the 'take rate' could be determined, defined as ration of lesions compared to the number of tissues injected. 

R2. Taking the Reviewers’ advice, we have added detailed method for endometriosis induction including the amount of initially injected endometrial tissues.

      While some tissue fragments diminish without growth, other tissue fragments survive to establish endometriotic lesions and grow their volume like human endometriosis. Therefore, it is difficult to calculate “take rate”.

C3. The authors provide a new method to increase visibility of the tissues in the recipient mice by either a genetically modified mice (mTmG) or by labeling the tissues with cy5 containing nanoparticles. However, did the authors tested whether it was indeed easier to find the lesions (eg by comparing tissue retrieval with and without the aid of fluorescence)?

R3. We published the endometriosis model using Pgrcre/+Rosa26mTmG/+ mice that is indeed easier to find the endometriotic lesions this year (Yoo JY et al, Nature Communications 2022). Current mouse models of endometriosis involve ovariectomy and E2 treatment to increase the size of endometriotic lesions that easy to distinguish endometriotic lesions from normal tissues. However, it is impractical for studies of physiological functions that require natural fluctuations in ovarian steroid hormones, such as fertility. Uncovering pathophysiological mechanisms of endometriosis with animal models requires easy identification of lesions to distinguish them from the surrounding normal tissues. With this in mind, we developed a mouse model of endometriosis using mT/mG reporters. Our mouse model alleviates the need to apply ovariectomy and E2 treatment to enlarge endometriotic lesions because fluorescence reporter genes allow us to visualize in vivo and in real-time endometriotic lesions like those found in humans. The mT/mG reporter is also useful to isolate pure GFP-positive endometriotic cells for molecular analysis.

C4. In addition, the cy5 labelled tissues are recovered one month later. However, the authors did not provide evidence of how long the nanoparticles or cy5 stain is able to remain over a period of time. Did the author perform longitudinal experiments to investigate the effect of time on the cy5 labeling?

R4. This is a very important comment. Although we have observed Cy5.5 signal at 3 month after endometriosis induction, the sample size and timepoint are not enough to present in the manuscript. Therefore, we have not included our data to the manuscript. We should perform longitudinal experiments to investigate the effect of time on the Cy5.5 labeling next time.

C5. Can cy5 labeling be readily analysed in ob/ob recipients with regard to adipose tissue? 

R5. Yes, understanding the pathophysiology of endometriosis with mouse models requires easy assays to distinguish endometriotic lesions from surrounding normal tissues. With this in mind, we developed an ex vivo labeling method of donor uterine tissues with Cy5.5 dye-doped silica nanoparticles.

C6. The analysis of endometriosis in the different types of mice is rather limited. Were other aspects of the pathogenesis investigated as well (eg neuroangiogenesis, inflammation, pain behaviour)? 

R6. Unfortunately, we do not have examined inflammation and pain behaviors in our mice with endometriosis. This is indeed a limitation of our study. Yes, we considered the possibility of examining molecular and phenotyping assays in our mice with endometriosis; however, this is technically challenging and requires synchronization of estrous cycle in female mice.

Reviewer 2 Report

This study by Kim et al. addresses a controversial field in endometriosis. While there is an inverse correlation between BMI and endometriosis, an increased BMI may lead to more severe forms of endometriosis. They used obesity mouse models to determine the role of leptin in the establishment of endometriosis lesions. They developed an ex vivo labeling method of donor uterine tissue using Cy5.5 dye-doped silica nanoparticles which was finally used in leptin deficiency mice (ob/ob) and leptin-receptor-deficiency mice (db/db).

In general the paper is very clear and concise. A few minor points:

1.    There are some typing errors and spelling mistakes. A careful revision with a native speaker will benefit the entire paper.

2.    In figure 4 and figure 5 the p-values and means do not fit together with the descriptions in the results section. Please correct this.

3.    Styrer at al 2008 (PMID: 17962343) performed similar experiments with leptin-receptor-deficiency mice. However, this has not been discussed.

4.    If the authors state in the heading that leptin stimulates establishment of endometriosis lesions in mouse models it would be of interest if a leptin peptide receptor antagonist (pegylated leptin peptide antagonist) would lead to reduction of ectopic lesions in the group of mice with diet-induced obesity too.

Author Response

This study by Kim et al. addresses a controversial field in endometriosis. While there is an inverse correlation between BMI and endometriosis, an increased BMI may lead to more severe forms of endometriosis. They used obesity mouse models to determine the role of leptin in the establishment of endometriosis lesions. They developed an ex vivo labeling method of donor uterine tissue using Cy5.5 dye-doped silica nanoparticles which was finally used in leptin deficiency mice (ob/ob) and leptin-receptor-deficiency mice (db/db).

In general, the paper is very clear and concise. A few minor points:

We are pleased with the positive comment.

C1. There are some typing errors and spelling mistakes. A careful revision with a native speaker will benefit the entire paper.

R1. We apologize and typing errors and spelling mistakes have been corrected. The manuscript has been proofread by all authors as well as experienced native research support staffs.

C2. In figure 4 and figure 5 the p-values and means do not fit together with the descriptions in the results section. Please correct this.

R2. We apologize the errors and have corrected the p-values in the manuscript.

C3. Styrer at al 2008 (PMID: 17962343) performed similar experiments with leptin-receptor-deficiency mice. However, this has not been discussed.

R3. We thank the reviewers for their time and effects in carefully reviewing the manuscript. This is a very important comment, and we agree with the reviewer. Although we mentioned the article in that regard, we now realize that that discussion was not made in full. Taking Reviewer’s suggestion, we have made changes in discussion.

C4. If the authors state in the heading that leptin stimulates establishment of endometriosis lesions in mouse models it would be of interest if a leptin peptide receptor antagonist (pegylated leptin peptide antagonist) would lead to reduction of ectopic lesions in the group of mice with diet-induced obesity too.

R4. This is a very insightful comment. In our experiment, we sought to show the necessity of leptin signaling in endometriosis development by removing leptin signaling with leptin deficient (ob/ob) mouse and leptin receptor deficient (db/db) mouse. Also, we demonstrated how supraphysiological levels of leptin in the peritoneum stimulates endometriosis development. We agree with the reviewer that it would be meaningful to demonstrate if blocking leptin signaling in mice with diet-induced obesity (with physiological levels of leptin) would also lead to reduction of ectopic lesions. This would be something that may be explored in future studies.